# Safety and Efficacy of Tyrosine Kinase Inhibitors in Immune Thrombocytopenic Purpura: A Systematic Review of Clinical Trials

**Muhammad Ashar Ali** [1,*,†], **Muhammad Yasir Anwar** [2,†], **Wajeeha Aiman** [3], **Gurneel Dhanesar** [1], **Zainab Omar** [4], **Mohammad Hamza** [5], **Maha Zafar** [6], **Harish Kumar Rengarajan** [1,‡] and **Michael Maroules** [7,‡]

1.  Department of Internal Medicine, St. Mary's and St. Clare's Hospitals, New York Medical College, Denville, NJ 07834, USA
2.  Department of Internal Medicine, BronxCare Hospital, Icahn School of Medicine, Bronx, NY 10457, USA
3.  Department of Internal Medicine, Saint Michael's Medical Center, New York Medical College, Newark, NJ 07102, USA
4.  Department of Pediatrics, Dubai Medical College for Girls, Dubai 20170, United Arab Emirates
5.  Department of Internal Medicine, Albany Medical Center Hospital, Albany, NY 12208, USA
6.  Department of Internal Medicine, Mercy Hospital Fort Smith, Arkansas College of Osteopathic Medicine, Fort Smith, AR 72903, USA
7.  Department of Hematology/Oncology, St. Mary's General Hospital, New York Medical College, Passaic, NJ 07102, USA
*   Correspondence: asharalianwar94@gmail.com
†   These authors contributed equally to this work.
‡   Senior authors.

**Abstract:** Immune thrombocytopenic purpura (ITP) is an acquired antibody or cell-mediated platelet damage or decreased platelet production. Steroids, IV immunoglobulins (IVIG), and Rho-anti-D antibodies are the commonly used initial treatments for ITP. However, many ITP patients either do not respond or do not maintain a response to initial therapy. Splenectomy, rituximab, and thrombomimetics are the commonly used second-line treatment. More treatment options include tyrosine kinases inhibitors (TKI), including spleen tyrosine kinase (Syk) and Bruton's tyrosine kinase (BTK) inhibitors. This review aims to assess the safety and efficacy of TKIs. **Methods:** Literature was searched on PubMed, Embase, WOS, and clinicaltrials.gov using keywords, "tyrosine kinase" and "idiopathic thrombocytopenic purpura". PRISMA guidelines were followed. **Results:** In total, 4 clinical trials were included with 255 adult patients with relapsed/refractory ITP. In all, 101 (39.6%) patients were treated with fostamatinib, 60 (23%) patients with rilzabrutinib, and 34 (13%) with HMPL-523. Patients treated with fostamatinib achieved a stable response (SR) and overall response (OR) in 18/101 (17.8%) and 43/101 (42.5%) of the patients, respectively, while SR and OR were achieved in 1/49 (2%) and 7/49 (14%) of the patients, respectively, in the placebo group. Patients treated with HMPL-523 (300 mg dose expansion) achieved an SR and OR in 5/20 (25%) and 11/20 (55%) of the patients, respectively, while SR and OR were achieved in 1/11 (9%) of the patients treated with the placebo. Patients treated with rilzabrutinib achieved an SR in 17/60 (28%) patients. Dizziness (1%), hypertension (2%), diarrhea (1%), and neutropenia (1%) were serious adverse events in fostamatinib patients. Rilzabrutinib or HMPL-523 patients did not require a dose reduction due to drug-related adverse effects. **Conclusions:** Rilzabrutinib, fostamatinib, and HMPL-523 were safe and effective in the treatment of relapsed/refractory ITP.

**Keywords:** tyrosine kinase inhibitors; immune thrombocytopenia; Bruton's tyrosine kinase; splenic tyrosine kinase; clinical trials

## 1. Introduction

Immune thrombocytopenic purpura (ITP) is an acquired form of platelet cell destruction due to an antibody or cell-mediated platelet damage or impaired platelet production,

usually causing a platelet count of $<100 \times 10^3/\text{mm}^3$ [1,2]. An annual incidence occurs in approximately 5 out of 100,000 per year in children and 2 in 100,000 adults per year, with a female predominance in patients younger than 70 years [3].

Most acute ITP cases, mostly in children, are mild and usually self-restricted. However, ITP symptoms vary widely among cases; in general, patients with platelet counts $> 50 \times 10^3/\text{mm}^3$ are asymptomatic and incidentally diagnosed. Bleeding is the major complication that may occur in around 60% of the patients, with a 13% incidence per year of fatal bleeding in >60 years old and 0.4%/year in <40 years old. The probability of severe bleeding is found to increase with increased patients' ages and with persistent platelet counts below $30 \times 10^3/\text{mm}^3$ [4]. The other often-reported symptoms by ITP patients were fatigue, a reduced health-related quality of life, and a greater risk for venous thromboembolism [5].

Multiple etiologies have been implicated in the pathogenesis of adult ITP. Studies reported idiopathic patterns and secondary ITP to heterogeneous bodily reactions against infections, medications, vaccines, and systemic illnesses [6]; subsequently, T-cell response dysregulation, leading to B-cell maturation into plasma cells. These plasma cells will produce autoantibodies that will act against platelets' variable surface proteins with resultant platelets' surface glycoproteins modification, platelets opsonization, phagocytosis by macrophages, and destruction in the spleen. Studies have also shown that the rule of these autoantibodies in delaying megakaryocyte maturation aggravates further platelet production reduction. An additional T-cytotoxic cell hyperactivation will follow with the consequent destruction of megakaryocytes and platelets [7]. These diverse mechanisms involved in ITP pathogenesis significantly impact ITP management, mainly aiming to stop and prevent severe bleeding with regard to the difficulty in estimating risks for bleeding. Inhibiting further platelet destruction and increasing the platelet count is another significant aspect of the management plan.

Steroids alone or in combination with IV immunoglobulins (IVIG) and anti-D therapy remain the primary line of ITP treatment through targeting B cells and, thus, the reduction of autoantibody production and another effect on antigen-presenting cells: the responsible cells for platelets' destruction and platelets' antigen presentation [8]. However, they have limited effects on the long-term resolution of chronic ITP symptoms and most of the patients require a second line of treatment [9]. The most-used second line of chronic ITP management is rituximab, another immunosuppressive medication with an anti-CD-20 activity, targeting and reducing the B cells number and maturation to autoantibodies producing plasma cells. However, studies showed controversy in its benefits as a long-term management modality [10].

Splenectomy is still considered the most effective choice of management in refractory ITP. Studies showed that up to two-thirds of splenectomy-treated patients had complete remission. The primary pathology among the remaining cases that experienced continuous post-splenectomy platelet destruction was mainly hepatic platelet destruction. However, surgical complications, the risk of life-threating infections, and thromboembolic events remain major challenges with splenectomy [11]. Thrombopoietin (TPO)-receptor agonists such as romiplostim and eltrombopag are the other line of management in ITP patients that stimulates megakaryocytes and platelet production. However, they failed to induce remission in a significant number of the patients, and major side effects of rebound thrombocytopenia upon the discontinuation of therapy and risk of thrombosis if platelets become too elevated [12].

Spleen tyrosine kinase (Syk, i.e., HMPL-523, fostamatinib) and Bruton tyrosine kinase (BTK) inhibitor (rilzabrutinib) have been active areas of research during the past recent years. BTKs are expressed by many cells and play a major role in antibody production, B-cell maturation, and Fc-receptor-mediated platelet phagocytosis. Similarly, Syk plays an important role in Fc-receptor-mediated signaling, leading to cell differentiation, proliferation, and the platelet phagocytosis process. Therefore, inhibiting these targets can reduce the progression of ITP. In this review, we will assess the safety and efficacy of tyrosine

kinase inhibitors (TKIs) in treating persistent ITP and a comparison with current treatment options. We will also discuss ongoing clinical trials and the need of future clinical trials.

## 2. Materials and Methods

Cochrane [13] and PRISMA [14] guidelines were followed by the authors in this systematic review.

### 2.1. Search Strategy

A comprehensive search was made on PubMed (Medline), Ovid Embase, Web of Science (WOS), and registry of clinicaltrials.gov with keywords, "tyrosine kinase" and "idiopathic thrombocytopenic purpura". The literature search was performed from the beginning of the data until 21 October 2022. The PICO framework was used to perform this literature search (Table S1) [15].

### 2.2. Inclusion and Exclusion of Articles

All the clinical trials providing safety (adverse effects) and efficacy (platelet response) data on TKIs inhibitors in AML were included. All the review articles, case reports, preclinical studies, and clinical studies irrelevant to TKI drugs or ITP were excluded. All the clinical trials without any safety or efficacy outcomes were also excluded.

### 2.3. Study Selection

Articles were screened by two authors (ZO and MH) and included based on pre-specified inclusion criteria. A third researcher (MAA) addressed the differences in screening.

### 2.4. Data Extraction

Two authors (GD and MYA) extracted the relevant data for the baseline characteristics of the included studies (treatment medication with dose, median age, splenectomy history, previous therapies, baseline platelet count), efficacy outcomes (stable response rate (SR), overall response (OR), modified stable response (MSR), and adverse events ($\geq$grade 3 adverse effects). A stable response was defined as platelets $\geq 50,000/\text{mm}^3 \geq 4$ of the six visits biweekly. An overall response was $\geq 1$ platelet count $\geq 50,000/\text{mm}^3$ in the first 12 weeks of treatment. A modified stable response was defined as $\geq 2$ platelet counts of $\geq 50,000/\text{mm}^3$, separated by a minimum of 5 days.

A primary outcome was a stable response. An overall response and a modified stable response were secondary outcomes for efficacy. An incidence of $\geq$grade 3 adverse effects were safety outcomes.

### 2.5. Risk of Bias (ROB) Assessment

ROB was conducted by using the Cochrane ROB-II tool by two researchers (WA and MZ) [16].

## 3. Results

From databases, 162 articles were identified from PubMed, 130 articles were identified from WOS, 179 from Embase, and 15 from the registry of clinicaltrials.gov. After careful screening of the articles, three randomized clinical trials (RCTs, *N* = 195) [17,18] and one non-randomized clinical trial (NRCT, *N* = 60) [19] were included; Figure 1.

### 3.1. Risk of Bias

The ROB was high in Kuter et al. 2022 as it was a single-arm phase I/II study and lacks randomization and blinding. There was some concern of bias in Yang et al. 2021 as insufficient data were available regarding the allocation of treatment regimen for each patient and blinding. Treatment outcomes, stable and overall response were not provided in all the subgroups of patients. The ROB was low in Bussel et al. 2018; Figure 2.

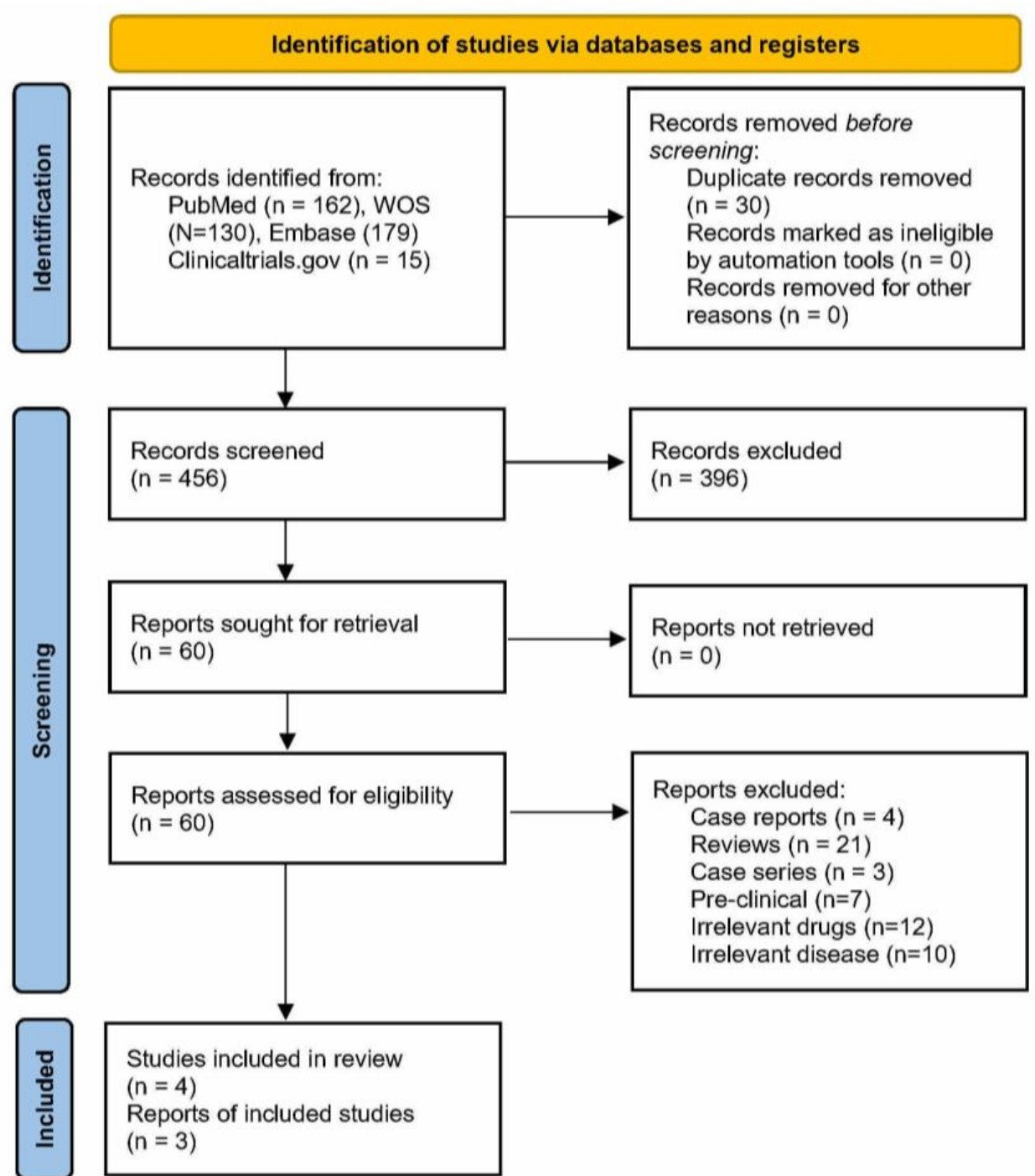

**Figure 1.** Prisma flowsheet of selection of articles.

### 3.2. Efficacy and Safety of TKI

In 4 clinical trials, 255 adult patients were treated with TKIs, 101 (39.6%) patients with fostamatinib, 60 (23%) patients with rilzabrutinib, and 34 (13%) with HMPL-523. In all, 43 (17%) patients had a relapse despite splenectomy, no prior splenectomy in 167 (65.4%) patients, and unknown splenectomy status in 45 (17.6%) patients. All the patients had prior steroid treatment. Baseline characteristics of the patients are given in Table 1.

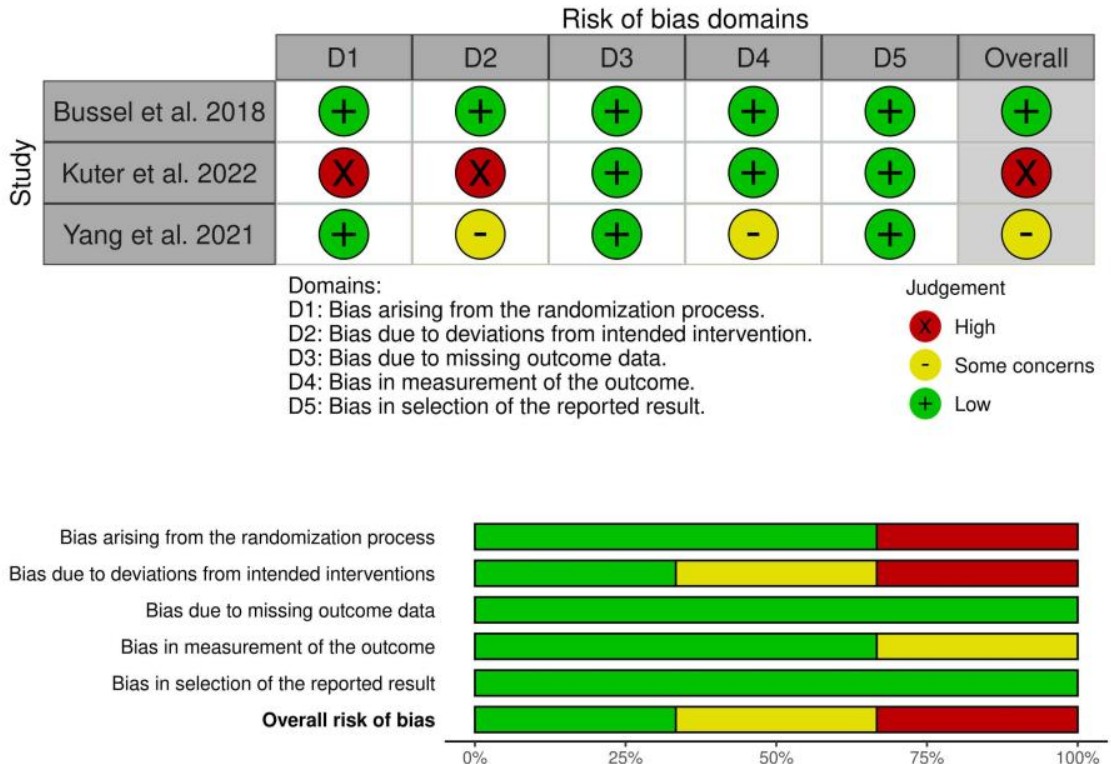

**Figure 2.** Risk of bias in included clinical trials.

**Table 1.** Baseline characteristics of patients in included clinical trials.

| Trial | NCT | Drug Therapy | Trial Phase | N | Median Age (Range) | ITP Classification n (%) | Prior Splenectomy n (%) | Most Common Prior Therapies | Mean Baseline Platelet Count (Range) |
|---|---|---|---|---|---|---|---|---|---|
| Bussel et al. 2018 [17] | NCT02076399 | Fostamatinib (100 mg BID) | Phase III randomized | 51 | 57 (20–88) | Persistent = 3 (6) Chronic = 48 (94) | 20 (39) | Corticosteroids 46 (90), IVIg, or IV Anti-D 33 (65); Thrombopoietic Agents 27 (53); Immunosuppressants 22 (43) | 16,202 (1000–51,000) |
| | | Placebo | | 25 | 57 (26–77) | Persistent = 3 (12) Chronic = 22 (88) | 10 (40) | Corticosteroids 25 (100), IVIg, or IV Anti-D 17 (68); Thrombopoietic Agents 15 (60); Immunosuppressants 12 (48) | 15,844 (1000–48,000) |
| | NCT02076412 | Fostamatinib (100 mg BID) | Phase III randomized | 50 | 50 (21–82) | Persistent = 3 (6) Chronic = 47 (94) | 14 (28) | Corticosteroids 48 (96), IVIg, or IV Anti-D 19 (38); Thrombopoietic Agents 20 (40); Immunosuppressants 22 (44) | 15,900 (1000–33,000) |
| | | Placebo | | 24 | 50 (20–78) | Persistent = 1 (4) Chronic = 23 (96) | 9 (38) | Corticosteroids 22 (92), IVIg, or IV Anti-D 10 (42); Thrombopoietic Agents 10 (42); Immunosuppressants 10 (42) | 23,958 (1000–156,000) |
| Kuter et al. 2022 [19] | NCT03395210 | Rilzabrutinib (200–400 mg) | Phase I-II Non-randomized | 60 | 50 (19–74) | Chronic = 52 (40) | 15 (25) | Corticosteroids 55 (92), IVIg 26 (43), Thrombopoietin Agent 35 (58) | 15,000 (2000–33,000) |
| | | 400 twice daily subgroup | | 45 | 49 (19–74) | | 11 (24) | Glucocorticoids 42 (93), IVIG 21 (47), Thrombopoietin Agent 24 (53) | 15,000 (2000–33,000) |

**Table 1.** *Cont.*

| Trial | NCT | Drug Therapy | Trial Phase | N | Median Age (Range) | ITP Classification n (%) | Prior Splenec-tomy n (%) | Most Common Prior Therapies | Mean Baseline Platelet Count (Range) |
|-------|-----|--------------|-------------|---|--------------------|--------------------------|--------------------------|-----------------------------|--------------------------------------|
| Yang et al. 2021 [18] | NCT03951623 | HMPL-523 (100 mg QD) | Phase I b randomized | 6 | 34.5 (20–58) | Concomitant = 1 (16.7) Non-concomitant = 5 (83.3) | NA | Prior therapy 6 (100) | 10,000 (1000–25,000) |
| | | 200 mg QD | | 6 | 33 (18–65) | Concomitant = 1 (16.7) Non-concomitant = 5 (83.3) | | Prior therapy 6 (100) | 4500 (3000–22,000) |
| | | 300 mg QD | | 16 | 40 (24–62) | Concomitant = 9 (56.3) Non-commitment = 7 (43.8) | | Prior therapy 16 (100) | 7000 (1000–26,000) |
| | | 400 mg QD | | 6 | 43 (28–57) | Concomitant = 0 Non-concomitant= 6 (100) | | Prior therapy 6 (100) | 8000 (2000–29,000) |
| | | Placebo | | 11 | 49 (21–65) | Concomitant = 5 (45.5) Non-concomitant = 6 (54.5) | | Prior therapy 11 (100) | 19,000 (4000–32,100) |

Concomitant = Multiple ITP mechanisms involved (i.e., ITP destruction and decreased production), Persistent=between 3 to 12 months from diagnosis not reaching spontaneous remission or not maintaining complete response off therapy, Chronic=lasting for more than 12 months.

### 3.2.1. Efficacy of Fostamatinib and HMPL-523 (Syk)

In two phase III RCTs (N = 135) by Bussel et al. 2018, patients were previously treated with a median of three lines of treatment. Patients in the fostamatinib group achieved an OR and SR in 43/101 (42.5%) and 18/101 (17.8%) of the patients, respectively, while OR and SR were achieved in 7/49 (14%) and 1/49 (2%) of the patients, respectively, in the placebo group. In a phase Ib RCT study by Yang et al. 2021 (N = 60), patients treated with HMPL-523 (300 mg dose expansion) achieved an SR and OR in 5/20 (25%) and 11/20 (55%) of the patients, respectively, while OR and SR were achieved in 1/11 (9%) of the patients in the placebo group; Table 2.

### 3.2.2. Efficacy of Bruton's TKI (Rilzabrutinib)

In the phase I–II clinical trial by Kuter et al. (*N* = 60), patients were previously treated with a median of four lines of treatment. Patients treated with rilzabrutinib achieved an SR and MSR in 17/60 (28%) and 24/60 (40%) patients, respectively. The median time to platelet count of $>50 \times 10^3/mm^3$ was 11.5 days; Table 2.

### 3.2.3. Safety

On treatment with fostamatinib, dizziness, hypertension, neutropenia, and diarrhea were serious adverse events reported in 1%, 2%, 1%, and 1% of the patients, respectively. Dose reduction due to adverse events was reported in 9% of the patients treated with fostamatinib. On treatment with rilzabrutinib, grade 1/2 diarrhea, nausea, fatigue, and vomiting were reported in 37%, 35%, 20%, and 7% of the patients, respectively. No treatment-related grade 3 or 4 adverse effects were seen on patients on rilzabrutinib. On treatment with HMPL-523, no adverse effects led to a dose reduction. Low-grade increased ALT, LDH, bilirubin, lipids, and blood pressure were reported in 25%, 25%, 20%, 10%, and 10% of the patients, respectively, on treatment with 300 mg HMPL-523; Table 2.

**Table 2.** Efficacy and safety data of tyrosine kinase inhibitors in immune thrombocytopenia.

| Trial | Drug Therapy | Stable Response * | Modified Stable Response ‡ | Overall Response € | Any Serious Adverse Effect | Serious Diarrhea | Serious Hypertension | Serious Dizziness | Serious Neutropenia | Serious Nausea | Serious Fatigue | Serious Abdominal Pain/Distension |
|---|---|---|---|---|---|---|---|---|---|---|---|---|
| Bussel et al. 2018 [17] | Fostamatinib (100 mg BID) | 9/51 (18%) | NA | 19 (37%) | 16% (9% dose limiting) | 1% | 2% | 1% | 1% | 0% | 0% | 0% |
| | Placebo | 0/25 | NA | 2 (8%) | 15% (2% dose limiting) | 0 | 2% | 0 | 0 | 0% | 0% | 0% |
| | Fostamatinib (100 mg BID) | 9/50 (18%) | NA | 24 (48%) | 16% | 1% | 2% | 1% | 1% | 0% | 0% | 0% |
| | Placebo | 1/24 (4%) | NA | 5 (21%) | 15% | 0% | 2% | 0% | 0% | 0% | 0% | 0% |
| Kuter et al. 2022 [19] | Rilzabrutinib (200–400 mg) | 17/60 (28%) | 24/60 (40%) | NA | 8 (0 = treatment related or dose limiting) | 0% | NA | NA | NA | 0% | 0% | 0% |
| | Rilzabrutinib 400 twice daily subgroup | 14/45 (31%) | 18/45 (40%) | NA | 0 | 0% | NA | NA | NA | 0% | 0% | 0% |
| Yang et al. 2021 [18] | HMPL-523 (100 mg QD) | NA | NA | 3/6 (50%) | No dose-limiting toxicity | | | | | | | |
| | 200 mg QD | NA | NA | 2/6 (33%) | No dose-limiting toxicity | 2 (5.9%) £ | NA | 2 (5.9%) £ | 3 (8.8%) £ | NA | NA | 2 (5.9%) £ |
| | 300 mg QD | 5/16 (31%) | NA | 11/16 (68.8%) | No dose-limiting toxicity | | | | | | | |
| | 400 mg QD | NA | NA | 2/6 (33%) | No dose-limiting toxicity | | | | | | | |
| | Placebo | 1/11 (9%) | NA | 1/11 (9%) | No dose-limiting toxicity | NA | NA | NA | NA | NA | NA | NA |

* = Platelets $\geq 50 \times 10^3/\text{mm}^3 \geq 4$ of 6 biweekly visits. € = $\geq 1$ platelet count $\geq 50 \times 10^3/\text{mm}^3$ within the first 12 weeks of treatment. ‡ = $\geq 2$ consecutive platelet counts, separated by at least 5 days of $\geq 50 \times 10^3/\text{mm}^3$. £ = any grade side effect.

### 3.3. Ongoing Clinical Trials

Seven clinical trials are in progress on Bruton's tyrosine kinase inhibitors registered on clinicaltrials.gov, including three trials on orelabrutinib (*N* = 80), three trials on zanubrutinib (*N* = 310), one RCT (*N* = 224) on rilzabrutinib, one trial on barcitinib (*N* = 33), one trial on SKI-O-703 (*N* = 60), and one trial on fostamatinib (*N* = 20); Table 3.

**Table 3.** Ongoing clinical trials on TKIs for refractory ITP registered on clinicaltrials.gov.

| NCT | Drug | Phase | N | Population | Outcomes | End Date |
|---|---|---|---|---|---|---|
| **Bruton tyrosine kinase and Janus kinase inhibitors** | | | | | | |
| NCT04562766 | Rilzabrutinib | Phase III RCT | 224 | Persistent and chronic ITP | Efficacy and safety | 2025 |
| NCT05446831 | Baricitinib | Phase II | 33 | Steroid re-lapsed/refractory ITP | Efficacy/safety | 2023 |
| NCT05124028 | Orelabrutinib | Phase I/II | 10 | Primary ITP | Efficacy and safety | 2022 |
| NCT05020288 | Orelabrutinib | Phase II | 40 | Refractory ITP | Efficacy and safety | 2024 |
| NCT05232149 | Orelabrutinib | Phase II | 30 | Refractory ITP | Efficacy and safety | 2024 |
| NCT05279872 | Zanubrutinib | Phase I/II | 10 | Primary ITP | Efficacy and safety | 2022 |
| NCT05369377 | Zanubrutinib + eltrombopag | Open label, RCT, phase II | 150 | Refractory ITP | Efficacy and safety | 2025 |
| NCT05369364 | Zanubrutinib + Dexametha-sone | Open label, RCT, phase II | 150 | First line ITP | Efficacy and safety | 2025 |
| **Spleen tyrosine kinase inhibitors** | | | | | | |
| NCT04056195 | SKI-O-703 | Phase II, RCT | 60 | Refractory ITP | Efficacy and safety | 2022 |
| NCT05509582 | Fostamatinib | Phase II | 20 | Post-transplant ITP | Efficacy and safety | 2028 |
| ITP = immune thrombocytopenia | | | | | | |

## 4. Discussion

Fostamatinib is an orally bioavailable competitive inhibitor of the Syk catalytic domain, tested in the treatment of ITP in adults with no or inadequate response to a prior therapy. Syk is expressed on B cells, macrophages, T cells, and platelets. Syk is activated when the Fc gamma receptor binds to its ligand and leads to the phosphorylation of activation tyrosine-based motifs in the immunoreceptor. In macrophages, these motifs lead to cytoskeletal changes and phagocytosis of platelets, while in B cells, Syk may have a role in antibody formation [20].

Intestinal alkaline phosphatase converts fostamatinib to an active metabolite R406. The active metabolite can inhibit the Fc epsilon receptor RI and Fc gamma receptor on mast cells, as well as Syk-dependent signaling on B cells [21]. Other possible mechanisms may include the inhibition of Jak, Lck, and Flt-3 pathways [21,22]. FIT1 and FIT2 were the multicenter phase III RCTs conducted by authors in Bussel et al. 2018 [17]. Fostamatinib was able to produce a response in a significant number of the patients who did not respond to TPO therapy, rituximab, and/or splenectomy. On further analysis by authors in Bussel et al. 2018, the response rate with fostamatinib was relatively higher in younger patients and platelet counts of $15–30 \times 10^3/mm^3$ as compared to older patients and platelet counts of $<15 \times 10^3/mm^3$. The response rate in patients with antiplatelet antibodies had a higher response rate as compared to patients without detectable antibodies (36% vs. 9%). Additionally or within the first 12 weeks were higher in patients with fostamatinib as the second line of therapy, as compared to the third, fourth, or fifth lines of therapy (78%, 64%, 52%, 36%, respectively).

Common treatment-related adverse events were mild/moderate and included hypertension, diarrhea, nausea, and transaminase elevation. No bleeding-related serious adverse events were reported. Medical management with antimotility and antihypertensives were sufficient for the patients with hypertension and gastrointestinal adverse effects. In the long-term phase III 5-year extension study by Cooper et al. 2021 (*N* = 146) [23], no new toxicities were observed. Only one thromboembolic event was reported as a mild transient ischemic attack that is much less than expected in an ITP patient. Platelet counts of >50 × $10^3$/mm$^3$ were reported in 54% of the patients. Initial results of a real-world study of fostamatinib were presented in the annual meeting of ASH 2022 conducted by authors in Moezi et al. 2022 (*N* = 46) [24]. The median number of prior therapies was 2 in this study. Platelet counts of >30 × $10^3$/mm$^3$ and >50 × $10^3$/mm$^3$ were achieved by 31 (67.4%) and 26 (56.5%) of the patients, respectively.

Fostamatinib is available in the market at a cost of about 11,000 USD for 1 month, which is comparable to the price of other oral agents such as eltrombopag (around 10,000 USD) and injectable agent romiplostim (around 9000 USD). Therefore, patient preference and medical necessity are major determinants of which agent to use rather than cost-effectiveness [25].

HMPL-523 is an oral agent that reversibly inhibits Syk; thus, preventing the destruction of opsonized platelets and auto-antibody formation, which is still under development and just completed its Phase 1b trial in China (NCT03951623) [18]. In this trial, HMPL-253 was well tolerated with no serious adverse events at a dose of 400 mg daily with a documented efficacy against the placebo. The adverse events seen include derangements in the liver-function test, hyperlipidemia, elevated amylase levels, hypokalemia, dizziness, diarrhea, proteinuria, and hypertension. In another study conducted in healthy Australian males, HMPL-523 caused febrile illness and elevated lipase levels [26]. HMPL-523 needs a high-powered study and head-to-head trials with other treatments such as rituximab or splenectomy to establish its relative efficacy. A phase III trial NCT05029635 is in progress to further assess the safety and efficacy of HMPL-523 in patients with refractory ITP. It can be used as an alternative agent to fostamatinib for SYK inhibition [27,28]. The drug is not yet approved by FDA and is not available in the market for ITP.

Rilzabrutinib is an oral BTK inhibitor (BTKI) that plays an important role in B-cell function, maturation, differentiation, and antibody formation [29]. Among other pathways, BTK is involved in Fc gamma receptor signaling and involves decreased macrophage-mediated platelet destruction and auto-antibodies formation. It has covalent and non-covalent binding sites; therefore, it can bind with high potency and can have long binding time [29]. Rilzabrutinib has a rapid on rate and a slow off rate with >80% binding in an hour and maintains receptor occupancy for 24 h. Unbound rilzabrutinib rapidly clears out of the system within 6 h, limiting systemic toxicity. In preclinical studies, rilzabrutinib, unlike other BTK inhibitors ibrutinib, does not affect the PI3K-Akt pathway that was believed to be associated with adverse effects such as atrial arrhythmia [29]. Similarly, older generation BTKIs such as ibrutinib also inhibit other kinases, which leads to decreased collagen-mediated platelet aggregation and are associated with bleeding diatheses. No such adverse effects were seen with rilzabrutinib in the short-term follow up [30]. The phase I/II clinical trial (NCT03395210) by Kuter et al. 2022 [19] included highly refractory ITP patients with a median of four lines of prior treatments (glucocorticoid, rituximab, TPO, IVIG, fostamatinib, splenectomy). In total, 400 mg twice daily was the highest dose tested in the trial and was able to produce a response in 40% of the patients; in addition, it was well tolerated by most of the patients. Further trials and testing are needed for the long-term efficacy and durability of rilzabrutinib treatment in patients with relapsing and refractory immune thrombocytopenia. Rilzabrutinib is under consideration by FDA for approval based on these results. A pivotal LUNA 3 phase III RCT (NCT04562766) is underway to assess rilzabrutinib treatment efficacy and safety. Therefore, it is not available in the market for ITP; however, it can be found for other diseases such as pemphigus vulgaris and may become available soon after FDA approval.

Trials are in progress for newer agents targeting tyrosine kinases, such as baritinib, orelabrutinib, zanubrutinib, and SKI-O-703. Baricitinib is an oral agent that binds reversibly with JAK proteins leading to the inhibition of the JAK-STAT pathway involved in gene transcription and the downstream activation of inflammatory mediators involved in autoimmune responses [31]. An early-phase trial is in progress in China to assess the safety and efficacy in steroid relapsed/refractory ITP patients (NCT05446831). Orelabrutinib is a BTKI-like rilzabrutinib and is involved in multiple signaling pathways of adaptive or innate immunity. In a preclinical study by Yu et al. 2021, the expression of CD69 and the BCR signaling pathway CD86 was significantly reduced with orelabrutinib in polymorphonuclear cells separated from ITP and healthy human subjects. In mice, the administration of orelabrutinib significantly increased the platelet count. Therefore, it can be an option for ITP patients in the future and human trials are in progress in China (NCT05124028, NCT05020288, and NCT05232149). Zanubrutinib is also a BTKI-like rilzabrutinib and orelabrutinib. A case report showed encouraging results of zanubrutinib in a 15-year-old Chinese girl with severe unresponsive ITP with Evans' syndrome [32]. Trials are in progress in China on monotherapy, and the combination of zanubrutinib with TPO and steroids in refractory ITP patients or first-line therapy (NCT05279872, NCT05369377, NCT05369364). SKI-O-703 is an Syk inhibitor such as fostamatinib and a phase II RCT is in progress in refractory ITP patients (NCT04056195).

Commonly used second line treatments for ITP are TPO, rituximab, and splenectomy. Splenectomy increases the infection risk especially with encapsulated bacteria, with reports of sepsis in 2–7 per person–year in splenectomy patients as well as an increased risk of thromboembolism. Surgical complications and the possibility of spontaneous remission of ITP in the first year, splenectomy is only considered after one year of diagnosis of ITP [33]. TKIs can be a valuable alternative for these patients for early use given a lesser number of severe adverse effects and may delay or completely obliviate the need of a splenectomy in these patients. Similarly, the combination of TKIs with a splenectomy can also be tested in clinical trials to reduce the risk of relapse since TKIs were able to produce a response in patients relapsed/refractory to splenectomy.

Rituximab is an off-label treatment of ITP for years. An early trial on rituximab in combination with dexamethasone as a first-line therapy showed encouraging results [34]; or in the range of 40–70%, were reported in patients after 4 weeks of rituximab of 375 mg/m$^2$ and on 5 years of follow up; the response was sustained in about 21% of the patients [10,35,36]. In the long-term RCT study by Ghanima et al. 2015 [10], there was no significant long-term benefit reported with the use of rituximab as second line of therapy. Rituximab trials included patients who had a first line of treatment only as compared to trials on ITP that included patients who failed multiple lines of treatment. There is no direct study comparing rituximab with TKIs. An indirect comparison was performed by network meta-analysis on RCTs by authors in Laws et al. 2022 [37]. Fostamatinib was significantly more effective in the overall improvement of platelet count than rituximab regimens with an odds ratio of 0.11–0.2. The risk of infection and reactivation of hepatitis B infection with rituximab is also increased. Given the COVID-19 pandemic, the increased risk of infection and possible impairment of vaccine response in about the 6-month period after administration do not make rituximab an ideal choice in the current era [38].

TPO mimics endogenous TPO and improves platelet production by stimulating megakaryocyte maturation. Multiple adverse events, including bone marrow fibrosis, continuous stimulation of megakaryocytes, increased risk of thromboembolism, transaminitis, and severe rebound thrombocytopenia are associated with TPO. Along with the use of TKIs in patients relapsed/refractory to TPO therapy, TKIs can also be considered for use with TPO therapy. Concomitant use of TKIs with TPO may increase the response rate, the success rate of tapering of TPO by preventing rebound thrombocytopenia, and may decrease the risk of thromboembolic events [23]. There is an ongoing trial in China on the combination of zanubrutinib with TPO; however, more large-scale multicenter RCTs are needed to provide reliable evidence.

*Limitations*

The results for RCT on rilzabrutinib were not yet available and only the non-randomized phase I/II study was available. The trial on HMPL-523 was a relatively small-scale dose-finding study. Large-scale RCTs are needed to confirm the safety and effectiveness of HMPL-523. Long-term study results were only available for fostamatinib. RCTs were conducted in comparison with the placebo and no trials were conducted on the comparison or combination of TKIs with current treatment options such as splenectomy, TPO, or rituximab. Secondary etiology of immune thrombocytopenia, such as HIV, H. pylori, Hep B, and C were elaborated in the clinical trials. The status of infections such as H. pylori were not clarified in the clinical trials since ITP may resolve with the resolution of these infections.

## 5. Conclusions

Fostamatinib and HMPL-523 (Syk inhibitors) were more effective than the placebo and were well tolerated by most of the patients with severe resistant ITP. In the early-phase clinical trial, rilzabrutinib oral (BTKI) was safe and well tolerated by most of the patients with ITP and was effective in patients with multidrug-resistant ITP. Large-scale phase III RCT is in progress to further assess the safety and efficacy of rilzabrutinib and HMPL-523 in comparison with the placebo. More novel TKI agents, such as orelabrutinib, zanubrutinib, baricitinib, and SKI-O-703, are under investigation in clinical trials. More large-scale multicenter RCTs are needed to assess the safety and effectiveness of TKIs in primary or secondary ITP.

**Supplementary Materials:** The following supporting information can be downloaded at: https://www.mdpi.com/article/10.3390/jox13010005/s1, Table S1. Search strategy in this article.

**Author Contributions:** Conceptualization, M.A.A. and M.Y.A.; methodology, M.A.A. and W.A.; software, W.A.; validation, G.D., Z.O. and M.H.; formal analysis, M.Z.; investigation, M.A.A. and W.A.; resources, M.A.A. and M.M.; data curation, G.D. and M.H.; writing—original draft preparation, M.A.A., G.D., Z.O., M.Z. and M.H.; writing—review and editing, H.K.R. and M.M.; visualization, H.K.R.; supervision, M.M. All authors have read and agreed to the published version of the manuscript.

**Funding:** This research received no external funding.

**Acknowledgments:** 

**Conflicts of Interest:** All the authors declare no conflict of interest.

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
