# Peer review of "Safety and Efficacy of Tyrosine Kinase Inhibitors in Immune Thrombocytopenic Purpura: A Systematic Review of Clinical Trials"

_jox, doi:10.3390/jox13010005_

Round 1
Reviewer 1 Report
This review article was a well-written review of four clinical trials on TKIs of ITP. I found the layout and sections of the article to be in order. I believe that the review article is nearly ready for publication. I cross checked some of the refrences with the citation and the text, and they seemed to be good.
A couple of critiques:
Figrue 1 text (in the figure iteself) is hard to read. The graphic shold have higher resolution.
It would be nice if table 2 fit on one page
247. Add a space after 24.
252 "No" instead of "Nos"
246 "400mg" The spacing should be consistent between the number and the units. In line 231, there is "400 mg."
Author Response
We appreciate the reviewers for taking out the time to review our manuscript thoroughly and giving us feedback. The reviewer’s valuable comments have helped us a lot in improving our manuscript. We have carefully pondered over all the comments and have made pertinent changes to our manuscript.
Responses to the individual comments are as under:
Figrue 1 text (in the figure iteself) is hard to read. The graphic shold have higher resolution.
Response: We have now added new a new image of figure 1 and is now readable. Thanks for pointing that out.
It would be nice if table 2 fit on one page.
Response: We have made some edits to fit the table into one page.
247. Add a space after 24.
Response: Space added after 24.
252 "No" instead of "Nos"
Response: Spelling error corrected. Thanks.
246 "400mg" The spacing should be consistent between the number and the units. In line 231, there is "400 mg."
Response: Changed 400 mg to 400mg. Now consistent as 400mg throughout the manuscript. Thanks.
Reviewer 2 Report
In this article, the authors provide a systematic review of studies reporting the use of tyrosine kinase inhibitors in the treatment of chronic ITP. Tyrosine kinase inhibitors are promising novel drugs for patients who are refractory to first and second line treatments.
The authors should also discuss the cost and availability of the drugs.
Minor corrections:
Page 1 Line 3, Line 19. Should read 'Immune thrombocytopenia' or 'immune thrombocytopenic purpura'
Page 7. 'non-commıtment' please correct
Page 6 & 7, Table 1. Please explain the ITP classifications in the foot note. (Persistent, Chronic, Concomitant, non-concomitant)
Table 3 'Splenic' should read as 'Spleen'
Page 9 Line 167, 168 please use the same unit for platelet count through out the manuscript
Author Response
We appreciate the reviewers for taking out the time to review our manuscript thoroughly and giving us feedback. The reviewer’s valuable comments have helped us a lot in improving our manuscript. We have carefully pondered over all the comments and have made pertinent changes to our manuscript.
Responses to the individual comments are as under:
Comment 1: The authors should also discuss the cost and availability of the drugs.
Response: Thanks for pointing out the missing information. We have added the costs and availability of drugs in the discussion section. (Page 11)
Minor corrections:
Page 1 Line 3, Line 19. Should read 'Immune thrombocytopenia' or 'immune thrombocytopenic purpura'
Response: Thanks for pointing that out. We have edited it accordingly.
Page 7. 'non-commıtment' please correct
Response: Thanks for pointing that out. We have edited it accordingly.
Page 6 & 7, Table 1. Please explain the ITP classifications in the foot note. (Persistent, Chronic, Concomitant, non-concomitant)
Response: We have now added it to the footnote of table 1. Thanks for pointing out the ambiguity.
Table 3 'Splenic' should read as 'Spleen'
Response: Thanks for pointing that out. We have edited it accordingly
Page 9 Line 167, 168 please use the same unit for platelet count through out the manuscript
Response: Thanks for pointing that out. We have edited it accordingly and platelet count units are now consistent throughout the manuscript.